# Sequence Image Interpolation via Separable Convolution Network

**Xing Jin** [1,2], **Ping Tang** [1,*], **Thomas Houet** [3], **Thomas Corpetti** [3], **Emilien Gence Alvarez-Vanhard** [3] **and Zheng Zhang** [1]

1   Aerospace Information Research Institute, Chinese Academy of Sciences, Beijing 100094, China; jinxing@radi.ac.cn (X.J.); zhangzheng@aircas.ac.cn (Z.Z.)
2   School of Electronic, Electrical and Communication Engineering, University of Chinese Academy of Sciences, Beijing 100049, China
3   CNRS UMR 6554 LETG, Université de Rennes 2, Place du recteur Henri le Moal, 35000 Rennes, France; thomas.houet@univ-rennes2.fr (T.H.); thomas.corpetti@univ-rennes2.fr (T.C.); emilien.alvarez-vanhard@univ-rennes2.fr (E.G.A.-V.)
*   Correspondence: tangping@aircas.ac.cn; Tel.: +86-139-1092-9397

**Abstract:** Remote-sensing time-series data are significant for global environmental change research and a better understanding of the Earth. However, remote-sensing acquisitions often provide sparse time series due to sensor resolution limitations and environmental factors, such as cloud noise for optical data. Image interpolation is the method that is often used to deal with this issue. This paper considers the deep learning method to learn the complex mapping of an interpolated intermediate image from predecessor and successor images, called separable convolution network for sequence image interpolation. The separable convolution network uses a separable 1D convolution kernel instead of 2D kernels to capture the spatial characteristics of input sequence images and then is trained end-to-end using sequence images. Our experiments, which were performed with unmanned aerial vehicle (UAV) and Landsat-8 datasets, show that the method is effective to produce high-quality time-series interpolated images, and the data-driven deep model can better simulate complex and diverse nonlinear image data information.

**Keywords:** sequence image interpolation; separable convolution network; separable convolution kernel; UAV dataset; Landsat-8 dataset

## 1. Introduction

Remote-sensing time-series data are an important part of big earth observation data. As standard spatiotemporal spectral data, remote-sensing time-series data can be applied to research and applications in global changes, such as vegetation phenology changes, land-surface parameter relationships, and land degradation. The value and successful application of remote-sensing time-series data are significant for earth science to expand the growth to a deeper level and to better understand the Earth [1,2].

Time-series analysis usually requires the data to be dense and has equal time intervals to facilitate the process. However, remote-sensing acquisitions often provide sparse time series due to sensor resolution limitations and environmental factors [3], such as cloud noise for optical data.

A conventional method to solve missing data is 1D data interpolation, as is usually done for moderate-resolution imaging spectroradiometer (MODIS) data sequences with the following processing characteristics. The interpolation method is essentially based on a 1D sequence in the time dimension. The sequence is relatively long. This method is not suitable for high-spatial-resolution sequence images containing fine spatial pattern information. Remote-sensing sequence images are a kind of short-range complex 2D data [4];

these sequences are rich in spatial information that must be considered during interpolation. Due to the limitation of sequence length, it is difficult to interpolate 2D images in the same way that an interpolation is applied on 1D data.

The simplest interpolation way is linear interpolation; however, because the land cover of the coverage area changes with time and these changes can show complex phenological dynamics, the simple and uniform weighted linear method cannot meet the requirements of spectral fidelity of interpolated images [5].

This paper was inspired by video frame interpolation and applies the idea to the remote-sensing field. The difference is that video frame interpolation focuses on estimating inter-frame motion, while remote-sensing sequence image interpolation focuses on estimating inter-scene spectral transformation.

Niklaus et al. [6] employ a deep fully convolutional neural network to estimate spatially adaptive 2D or separable 1D convolution kernels for each output pixel and convolves input frames with them to render the intermediate frame. The convolution kernel captures both local motion between input frames and the coefficients for pixel synthesis. The key to making this convolution approach practical is to use 1D kernels to approximate full 2D ones. The use of 1D kernels significantly reduces the number of kernel parameters and enables full-frame synthesis.

This paper uses the same idea, and also employs a contraction–expansion of the deep fully convolutional neural network to estimate spatially adaptive separable 1D convolution kernels for each output pixel; the convolution kernel captures the local inter-scene spectral transformation coefficients for pixel synthesis.

To the best of our knowledge, this is one of the first attempts to use the prototype of a fully convolutional neural network to estimate inter-scene spectral transformation for the interpolation of remote-sensing sequence images. The major novelty of this paper can be summarized as follows:

- We use adaptive data-driven model for inter-scene spectral transformation of remote-sensing images, and provide a robust interpolation approach for making up the missing remote-sensing images.
- We verify, by experiments, the possibility of simulating missing remote-sensing image scenes of specified acquisition times and remote-sensing sequences at equal time intervals using the proposed data-driven spatially adaptive convolution network. This allows the processing of remote-sensing sequences to be carried out under a unified framework, instead of requiring different processing logic for each sequence due to different time intervals.

This paper shows that the data-driven model can better simulate complex and diverse nonlinear inter-scene spectral transformation, then get the inter-scene interpolated image based on this data-driven model. High-quality time series of interpolated images can be produced by the same approach. This enriches the research and development of the remote-sensing field.

The rest of the paper is organized as follows. Section 2 reviews related studies regarding remote-sensing time-series data interpolation. Experimental datasets and the proposed separable convolution network are described in Section 3. Section 4 presents the experiments and results, including visual comparisons and quantitative evaluation with other methods. Section 5 discusses the influence of hyperparameters on the interpolated result. Section 6 concludes the paper.

## 2. Related Studies

Remote-sensing data interpolation methods can be divided into two major types according to known spatial and temporal neighborhood data. The first is to establish a suitable spatial interpolation model based on the spatial relationship between spatial neighborhood data. The second is to establish a corresponding time-series interpolation model based on the time characteristics between temporal neighborhood data.

Seaquist et al. [7] used the ordinary kriging (OK) method to improve the accuracy of a normalized difference vegetable index (NDVI) maximum value composite (MVC) synthesized time series. Mercedes et al. [8] used a spatial interpolation method to interpolate leaf-area index (LAI) data. Shrutilipi et al. [9] compared the accuracy of remote-sensing data interpolated by different spatial interpolation methods and concluded that the accuracy of OK was better than inverse distance weight (IDW). The above methods can use the spatial information of remote-sensing images for spatial interpolation, and cannot use time-series information for temporal interpolation.

Zhou et al. [10] used NDVI data of the MODIS satellite to conduct simulation experiments and evaluate the Savitzky–Golay (SG) filtering [11] and harmonic analysis of time-series (HANTS) model [12] refactoring effect at different time intervals. According to the daily harmonic changes of land surface temperature (LST), Crosson et al. [13] used the LST data of MODIS Terra and Aqua to repair missing LST points by harmonic analysis. The above methods provide better 1D data fitting for the interpolation of time-series data at different time intervals, and are not suitable for the interpolation of high-dimensional time-series data (remote-sensing sequence images).

Recently, the emergence of the enhanced spatial and temporal adaptive reflectance fusion model (ESTARTFM) [14], spatial and temporal adaptive reflectance fusion model (STARTFM) [15], and global dense feature fusion convolutional network [16] has provided ideas for research on time-series image interpolation. These models can obtain high temporal and spatial resolution fusion data, but they cannot elaborate on the spatiotemporal evolution of sequence images.

It is not enough to consider remote-sensing data interpolation only from the temporal or spatial dimension. Our proposed separable convolution network combines the temporal neighborhood of predecessor and successor images and the spatial neighborhood to consider the interpolation of scene-based remote-sensing sequence images. It does not rely on other high-temporal-resolution remote-sensing data, and only interpolates based on the sequence itself. This provides a new idea for the interpolation of remote-sensing data.

## 3. Materials and Methods

### 3.1. Datasets

This paper uses two datasets (UAV and Landsat-8) for experiments. Figure 1 shows the location of the UAV dataset: the Sougéal marsh (western France, 48.52°N, 1.53°W), which is part of the long-term socio-ecological research (LTSER) site Zone Atelier Armorique. This site is a large flooded grassland of 174 ha located in the floodplain of the Couesnon River, upstream of Mont-Saint-Michel Bay [17]. The projection type is France Lambert-93. The spatial resolution is 0.02 m. The number of bands is 4: green, red, red-edge, and near-infrared. Figure 2 shows the location of the Landsat-8 dataset, which located in the southeast of Gansu Province (Path: 129, Row: 37, 33.44°N, 105.06°E). The projection type is Universal Transverse Mercator (UTM). The spatial resolution is 30 m. The number of bands is 7: coastal, blue, green, red, near-infrared, short-wave infrared-1, and short-wave infrared-2.

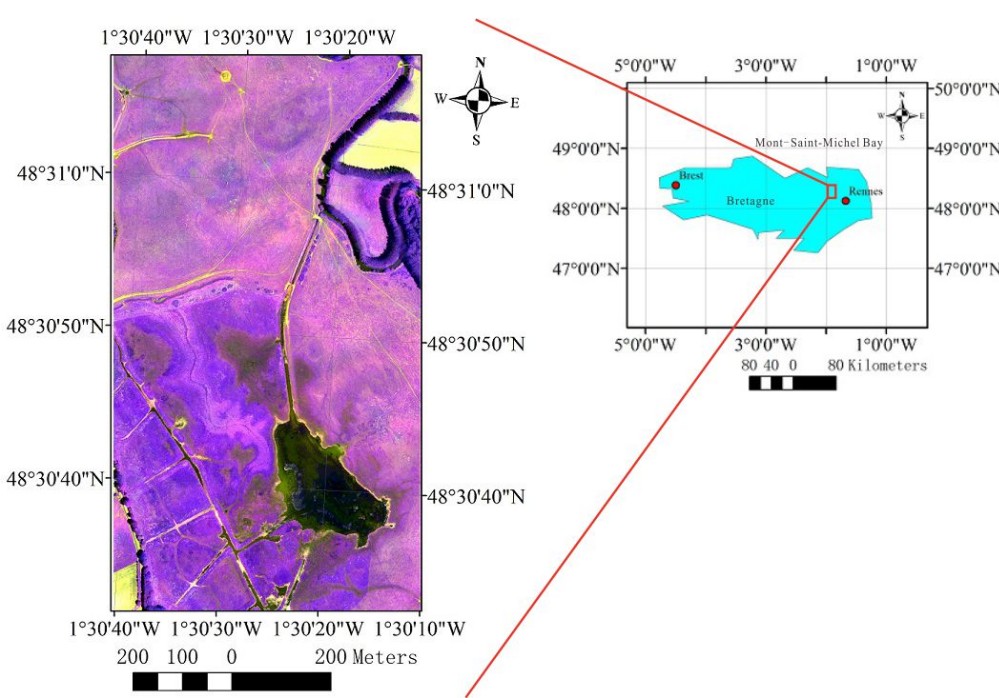

**Figure 1.** Location of unmanned aerial vehicle (UAV) dataset.

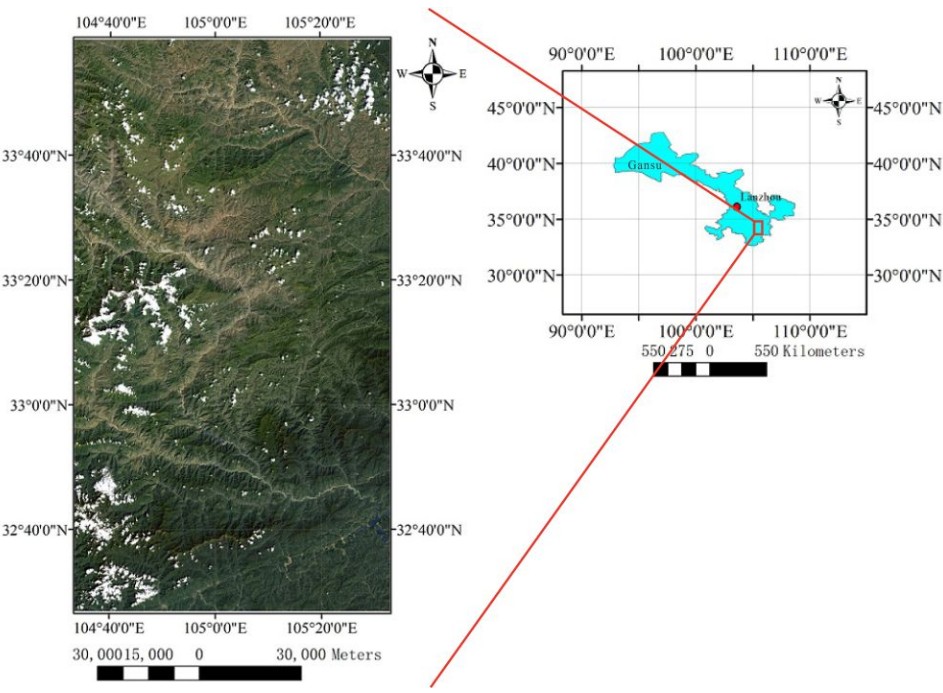

**Figure 2.** Location of Landsat-8 dataset.

### 3.2. Theoretical Model

Given two images $I_{t_1}$ and $I_{t_2}$ temporally in a sequence, it is reasonable to assume the middle image $I_{\text{estimated}}$ between images $I_{t_1}$ and $I_{t_2}$ could be estimated by Equation (1):

$$I_{estimated} = b_1(x,y) * K_1(x,y) + b_2(x,y) * K_2(x,y) \tag{1}$$

where $b_1(x,y)$ and $b_2(x,y)$ are the patches centered at (x, y) in $I_{t_1}$ and $I_{t_2}$, and $K_1(x, y)$ and $K_2(x, y)$ are a pair of 2D convolution kernels; note that $*$ denotes a local convolution operation. The pixel-dependent kernels $K_1$ and $K_2$ capture both motion and re-sampling information required for interpolation. The 2D kernels, $K_1$ and $K_2$, could be approximated

by a pair of 1D kernels. That is, $K_1$ could be approximated as $k_{1,v} * k_{1,h}$ and $K_2$ could be approximated as $k_{2,v} * k_{2,h}$. Under this assumption, the main task is to estimate each separable 1D kernel parameter $k_{1,v}$, $k_{1,h}$, $k_{2,v}$, $k_{2,h}$.

Furthermore, the changes between $I_{estimated}$ and the changes of $I_{t_1}$ and $I_{t_2}$ over time are considered to be nonlinear. The 1D kernel parameter functions $k_{1,v}$, $k_{1,h}$, $k_{2,v}$, $k_{2,h}$ can be assumed, and both are nonlinear mappings that can be represented by convolutional neural networks. Without loss of generality, we assume that the four 1D kernels have a tightly supported set. We apply the kernels to each of the multispectral channels to synthesize the output pixel.

### 3.3. Architecture of the Model

The architecture of the model is shown in Figure 3; the separable convolution network consists of a contracting part and an expanding part. The contracting part is used to extract features of training samples, and the expanding part is used to recover the extracted features.

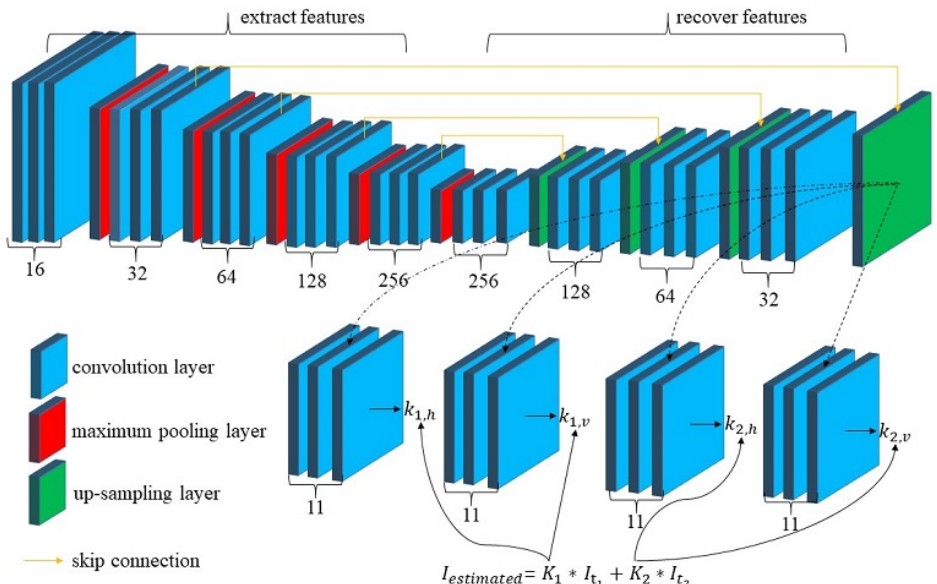

**Figure 3.** Overview of our separable convolution network architecture.

The extracting part mainly contains five convolution layers and five pooling layers. The number of filters in each convolution layer is 16, 32, 64, 128, and 256. Stacks of 3 × 3 convolution with rectified linear unit (ReLu) are used in each convolution layer. Maximum pooling is used in each pooling layer. The expanding part mainly contains four deconvolution layers and four upsampling layers. The number of filters in each deconvolution layer is 256, 128, 64, and 32. The upsampling layers can be executed in various ways, such as nearest-neighbor, bi-linear interpolation, and cubic convolution interpolation [18–20]. Furthermore, we utilize skip connection [21,22] to let upsampling layers incorporate features from the contracting part of the separable convolution network. To estimate four sets of 1D kernels, we direct the feature information in the last expansion layer into four sub-networks, with each sub-network evaluating one kernel.

In our experiments, the default image block size is 125 × 125 pixels and the separable convolution kernel size is 11 × 11 pixels. Our approach shares the different convolution kernels to each of the input channels.

*3.4. Loss Functions*

Our research uses two types of loss function, $\ell_{mse}$ loss and $\ell_c$ loss, which measure the difference between an interpolated image $I_{estimated}$ and corresponding reference image $I_{gt}$. The first loss function is $\ell_{mse}$ norm based on pixel difference and is defined in Equation (2):

$$\ell_{mse} = \frac{1}{n}\Sigma(||I_{estimated} - I_{gt}||_2) \tag{2}$$

The second loss function is $\ell_c$ norm based on the combination of feature difference and pixel difference and is defined in Equation (3):

$$\ell_c = \ell_{mse} + ||\varphi(I_{estimated}) - \varphi(I_{gt})||_2 \tag{3}$$

where $\varphi$ extracts features from an image. We tried to use feature extractors like visual geometry group (VGG-19) [23]. During feature extraction, interpolated result and reference image are intercepted to the 10th layer of the VGG-19 network, which has a total of 16 layers. The extracted feature is usually based on high-level features of input images, and it can increase the high-frequency components of the interpolated result. To check their result, we used two versions of our convolution model. For the first and second loss, we used $\ell_{mse}$ loss and $\ell_c$ loss for simplicity in this paper.

*3.5. Evaluation Indicator*

Our research uses two evaluation indicators to evaluate the quality of the interpolated result: root mean square error (RMSE) and entropy function. RMSE measures the pixel error between an interpolated image $I_{estimated}$ and corresponding reference image $I_{gt}$, as defined in Equation (4):

$$RMSE = \sqrt{\frac{1}{n}\Sigma(||I_{estimated} - I_{gt}||_2)} \tag{4}$$

The entropy function based on statistical features is an important indicator to measure the richness of image information. The information amount of an image *I* is measured by the information entropy *D(I)*, as defined in Equation (5):

$$D(I) = -\sum_{i=0}^{L-1} P_i ln(P_i) \tag{5}$$

where $P_i$ is the probability of a pixel with a gray value of *i* in image, and *L* is the total number of gray levels (*L* = 256). According to Shannon's information theory [24], there is the most information when there is maximum entropy. Generally speaking, the larger the *D(I)*, the clearer the image. The benefit of using entropy over RMSE is that entropy can capture the amount of information in the image, and the detailed information of the image can be reflected indirectly through entropy.

## 4. Experiments and Results

*4.1. Training Strategy*

We take three scenes composed of one sequence from the dataset as examples, and select the first two scenes as the input and the last scene as the output to train the model. Each sequence was trained to get one network model.

The size of scenes in sequence in both UAV and Landsat-8 datasets was 3100 × 5650. To get enough training samples, all scenes in a sequence were aligned and divided into three regions: a training region with a size of 2000 × 2000, accounting for 22.8% of the entire scene; a validation region with a size of 2000 × 2000, also 22.8% of the scene; and the rest of the scene, used as the testing sample, accounting for 55.4% of the entire scene. All

three regions were continuously cropped as a block with a size of 125 × 125, and there was overlap when cropping.

The optimizer used in the training was Adamax with $\beta_1 = 0.9$, $\beta_2 = 0.99$, and a learning rate of 1e-3. Compared to other network optimizers, Adamax could achieve better convergence of the model [25].

### 4.2. Testing Strategy

This paper mainly conducted three sets of the experiment. Each set had an experimental purpose and corresponding data. The first set was mainly designed to verify the effectiveness of our proposed method; this experiment was implemented within the scene of sequences. The second set was a generalized application in the time dimension, and this experiment was implemented between two sequences. The third set of the experiment was mainly to generate missing images in different time series using the proposed method, and this was implemented among multiple sequences. The corresponding experimental data are described in the following paragraphs in detail.

Before describing the experimental data in every experiment, we will first introduce some symbols used below. I represents image scene in sequence, and $I_{t_3}$ represents the image acquired at time $t_3$. The mapping model generated by $\ell_c$ loss is expressed as $f^{\ell c}_{[I_{t_1}, I_{t_2}, I_{t_3}]}$, and the mapping model generated by $\ell_{mse}$ loss is expressed as $f^{\ell mse}_{[I_{t_1}, I_{t_2}, I_{t_3}]}$, where $[I_{t_1}, I_{t_2}, I_{t_3}]$ represents training image triples, $I_{t_1}$ and $I_{t_2}$ represent the training image pairs, $I_{t_3}$ represents the reference image, $t_1$ and $t_2$ represents the month of training image pairs acquired, and $t_3$ represents the month of the reference image acquired. $f^{\ell c}_{[I_{t_1}, I_{t_2}, I_{t_3}]}(I_{t_1}, I_{t_2})$ represents output image with mapping model $f^{\ell c}_{[I_{t_1}, I_{t_2}, I_{t_3}]}$ and input scenes $I_{t_1}$ and $I_{t_2}$, and $f^{\ell mse}_{[I_{t_1}, I_{t_2}, I_{t_3}]}(I_{t_1}, I_{t_2})$ has a similar meaning.

In the first set of the experiment, both the testing samples and the generated results were blocks within the scene of sequences, and this is called block interpolated results. The reference block was the real one there. Our proposed method and the method of Meyer et al. [26] were compared in this experiment. The method of Meyer et al. was extrapolated, the extrapolated result was expressed as $f_{[I_{t_1}, I_{t_2}]}(I_{t_1})$, where $f_{[I_{t_1}, I_{t_2}]}$ represents extrapolated mapping trained by training image pair $[I_{t_1}, I_{t_2}]$ with reference image $I_{t_2}$. $I_{t_1}$ in the brackets represents the input used to generate the extrapolated result.

Table 1 shows the sequences used in this experiment and the dates of all scenes acquired in them. Figure 4 shows the distribution of training blocks, testing blocks, and validation samples within all the scenes, where the area inside the red and green boxes indicate the training and testing blocks, and the others are validation samples.

**Table 1.** Name and date of experimental datasets in first set of experiment.

| Dataset | Image Names | Image Dates |
|---|---|---|
| UAV | $I_4$ | April 2019 |
| | $I_5$ | May 2019 |
| | $I_6$ | June 2019 |
| | $I_7$ | July 2019 |
| | $I_8$ | August 2019 |
| Landsat-8 | $I_4$ | April 2013 |
| | $I_7$ | July 2013 |
| | $I_9$ | September 2013 |
| | $I_{11}$ | November 2013 |
| | $I_{12}$ | December 2013 |

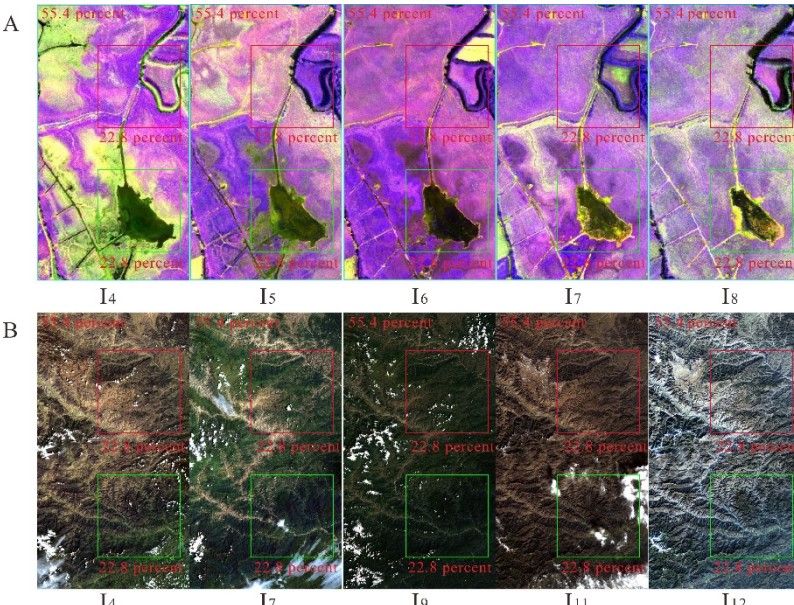

**Figure 4.** Distribution of training, testing, and validation samples in first set of experiment: (**A**) unmanned aerial vehicle (UAV) and (**B**) Landsat-8 images; areas inside red and green box and remainder of images show distribution of training, testing, and validation samples, respectively.

The second set of the experiment was implemented between two sequences. The mapping model was generated by one sequence and the application by an adjacent sequence, that is, the testing frames out of another sequence, and here the generated result is called the scene interpolated result. If the mapping model generated by $\ell_c$ loss was $f^{\ell c}_{[I_{t_1}, I_{t_2}, I_{t_3}]}$, the scene interpolated result was $f^{\ell c}_{[I_{t_1}, I_{t_2}, I_{t_3}]}(I'_{t_1}, I'_{t_2})$. If the mapping model generated by $\ell_{mse}$ loss was $f^{\ell mse}_{[I_{t_1}, I_{t_2}, I_{t_3}]}$, the scene interpolated result was $f^{\ell mse}_{[I_{t_1}, I_{t_2}, I_{t_3}]}(I'_{t_1}, I'_{t_2})$. $I'_{t_1}$ and $I_{t_1}$ represent images of the same place in month $t_1$. $I'_{t_2}$ and $I_{t_2}$ represents images of the same place in month $t_2$. Figure 5 shows the visual effect between training image $(I_{t_1}, I_{t_2})$ and testing image $(I'_{t_1}, I'_{t_2})$.

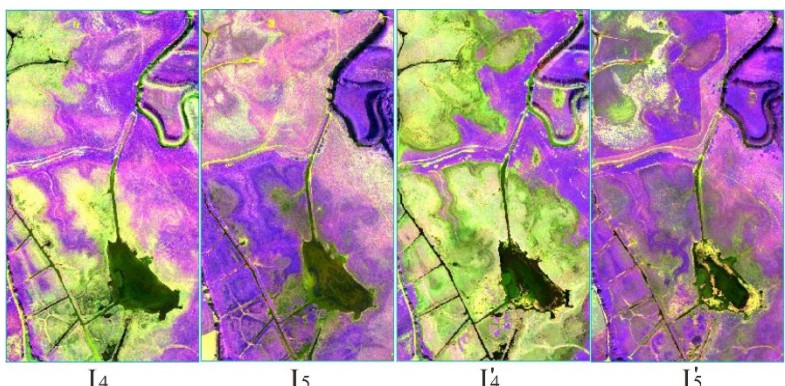

**Figure 5.** Visual effect of training and testing images in second set of experiment ($I_4$ and $I_5$ show visual effect of training image; $I'_4$ and $I'_5$ show visual effect of testing image).

The third set of the experiment was conducted with multiple sequences. The remote-sensing sequence here mainly reflected two aspects: (1) non-equidistant missing images in the same time series and (2) non-equidistant missing images of the same scene in different time series. It was difficult to find an analysis method to analyze these sequences in a unified and integrated manner. The number of sequences in a year may be relatively small, and the time interval between images uncertain, so there were non-equal time intervals; some were long and some were short. Figure 6A shows available UAV images

from 2017 to 2019 in this experiment. It is obvious there were many missing images for the frequency of one image per month. This experiment tried to generate those missing images. Figure 6B shows one strategy for generating missing data. The red points mark the first level interpolated result, in which training and testing images are both existing images; lines of the same color connect two testing images; green points mark the second level interpolated result, of which red point images are among training or testing images; cyan-blue points mark the third level interpolated result, of which green point images are among training or testing images. In this experiment, our method produced 19 scene images, and the network model was trained 19 times. The mapping model generated during each training was used to generate a new scene image. The training triplet images, testing images, and output images are listed in Table 2.

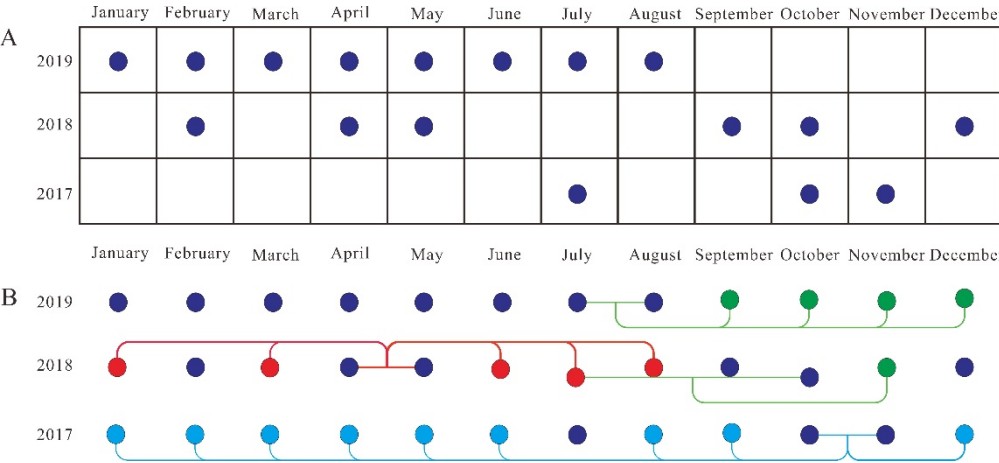

**Figure 6.** Strategy for generating missing data in third set of experiment: (**A**) available UAV images from 2017 to 2019; (**B**) one generation strategy. Red, green, and cyan-blue curves show lines of first, second, and third level of interpolated result.

**Table 2.** Training triplet images, testing images, and output images in third set of experiment.

| Color of Points | Training Triplet Images | Testing Images | Output Images |
|---|---|---|---|
| Red | April, May, January 2019 | April, May 2018 | January 2018 |
| | April, May, March 2019 | | March 2018 |
| | April, May, June 2019 | | June 2018 |
| | April, May, July 2019 | | July 2018 |
| | April, May, August 2019 | | August 2018 |
| Green | July, October, November 2017 | July, October 2018 | November 2018 |
| | July, August, September 2018 | July, August 2019 | September 2019 |
| | July, August, October 2018 | | October 2019 |
| | July, August, November 2018 | | November 2019 |
| | July, August, December 2018 | | December 2019 |
| Cyan-blue | October, November, January 2018 | October, November 2017 | January 2017 |
| | October, November, February 2018 | | February 2017 |
| | October, November, March 2018 | | March 2017 |
| | October, November, April 2018 | | April 2017 |
| | October, November, May 2018 | | May 2017 |
| | October, November, June 2018 | | June 2017 |
| | October, November, August 2018 | | August 2017 |
| | October, November, September 2018 | | September 2017 |
| | October, November, December 2018 | | December 2017 |

*4.3. Experimental Details*

Image blocking: Due to the limitation of the sequence image size, inputting the whole scene image (image size is 3100 × 5650) would cause the system to run out of memory. To address this issue, remote-sensing images needed to be processed in blocks during the experiment. The input image was divided into 25 blocks (each 620 × 1130), which could cover the entire image information and relieve memory pressure.

Time complexity: We used the Python machine learning library (PyTorch) to execute this separable convolution network. To improve computational efficiency, we organized our layer in computer unified device architecture (CUDA) that applies estimated 1D kernels. Our network was able to interpolate a 620 × 1130 image in 40 s. Obtaining the overall scene image (image size 3100 × 5650) took about 15 min under the acceleration of the graphics processing unit (GPU) [27].

*4.4. Results*

Table 3 shows the quantitative evaluation indicator between the block interpolated result and reference block using our proposed method in both datasets in the first set of the experiment. The table shows that the entropy value produced by using $\ell_c$ loss was higher than that produced by using $\ell_{mse}$ loss, and the RMSE [28] value produced by using $\ell_{mse}$ loss was lower than that produced by using $\ell_c$ loss. Table 4 shows a quantitative comparison between the block interpolated result and reference block using our proposed method and the method of Meyer et al. on both datasets. The table shows that the entropy value was higher and the RMSE value was lower using our method compared to the values produced using Meyer et al.'s method on both datasets. Figure 7 shows the visual effect and pixel error between the block interpolated result and reference block using different loss functions in both datasets and illustrates that using $\ell_{mse}$ loss led to visually blurry results, and using $\ell_c$ loss led to clear results with more high-frequency components [29–31] in our proposed method. Figure 8 shows the visual comparison and pixel error between the block interpolated result and the reference block using our method and the method of Meyer et al. on both datasets, and illustrates that the block interpolated result using our method was close to the spectral features of the reference block.

**Table 3.** Quantitative evaluation between block interpolated result and reference block using our proposed method on both datasets.

| Dataset | Interpolated Results (Block) | Reference Blocks | Entropy | RMSE (Pixel) |
|---|---|---|---|---|
| UAV | $f^{\ell mse}_{[I_4, I_5, I_6]}(I_4, I_5)$ | $I_6$ | 3.719 | 1.052 |
| | $f^{\ell c}_{[I_4, I_5, I_6]}(I_4, I_5)$ | | 3.723 | 1.077 |
| | $f^{\ell mse}_{[I_4, I_5, I_7]}(I_4, I_5)$ | $I_7$ | 3.441 | 1.070 |
| | $f^{\ell c}_{[I_4, I_5, I_7]}(I_4, I_5)$ | | 3.450 | 1.294 |
| | $f^{\ell mse}_{[I_4, I_5, I_8]}(I_4, I_5)$ | $I_8$ | 3.498 | 1.116 |
| | $f^{\ell c}_{[I_4, I_5, I_8]}(I_4, I_5)$ | | 3.508 | 1.429 |
| Landsat-8 | $f^{\ell mse}_{[I_4, I_7, I_9]}(I_4, I_7)$ | $I_9$ | 3.143 | 0.817 |
| | $f^{\ell c}_{[I_4, I_7, I_9]}(I_4, I_7)$ | | 3.145 | 1.112 |
| | $f^{\ell mse}_{[I_4, I_7, I_{11}]}(I_4, I_7)$ | $I_{11}$ | 3.842 | 1.233 |
| | $f^{\ell c}_{[I_4, I_7, I_{11}]}(I_4, I_7)$ | | 3.846 | 1.321 |
| | $f^{\ell mse}_{[I_4, I_7, I_{12}]}(I_4, I_7)$ | $I_{12}$ | 3.545 | 1.040 |
| | $f^{\ell c}_{[I_4, I_7, I_{12}]}(I_4, I_7)$ | | 3.550 | 1.476 |

**Table 4.** Quantitative comparison between block interpolated result and reference block using our proposed method and method of Meyer et al. on both datasets.

| Dataset | Interpolated Results (Block) | Reference Blocks | Entropy | RMSE (Pixel) |
|---|---|---|---|---|
| UAV | $f_{[I_4,I_5,I_6]}^{\ell mse}(I_4,I_5)$ | $I_6$ | 3.719 | 1.052 |
| | $f_{[I_4,I_6]}(I_4)$ | | 3.701 | 1.320 |
| | $f_{[I_4,I_5,I_7]}^{\ell mse}(I_4,I_5)$ | $I_7$ | 3.441 | 1.070 |
| | $f_{[I_4,I_7]}(I_4)$ | | 3.440 | 1.888 |
| | $f_{[I_4,I_5,I_8]}^{\ell mse}(I_4,I_5)$ | $I_8$ | 3.498 | 1.116 |
| | $f_{[I_4,I_8]}(I_4)$ | | 3.477 | 2.369 |
| Landsat-8 | $f_{[I_4,I_7,I_9]}^{\ell mse}(I_4,I_7)$ | $I_9$ | 3.143 | 0.817 |
| | $f_{[I_4,I_9]}(I_4)$ | | 3.125 | 1.550 |
| | $f_{[I_4,I_7,I_{11}]}^{\ell mse}(I_4,I_7)$ | $I_{11}$ | 3.842 | 1.233 |
| | $f_{[I_4,I_{11}]}(I_4)$ | | 3.572 | 1.957 |
| | $f_{[I_4,I_7,I_{12}]}^{\ell mse}(I_4,I_7)$ | $I_{12}$ | 3.545 | 1.040 |
| | $f_{[I_4,I_{12}]}(I_4)$ | | 3.541 | 1.769 |

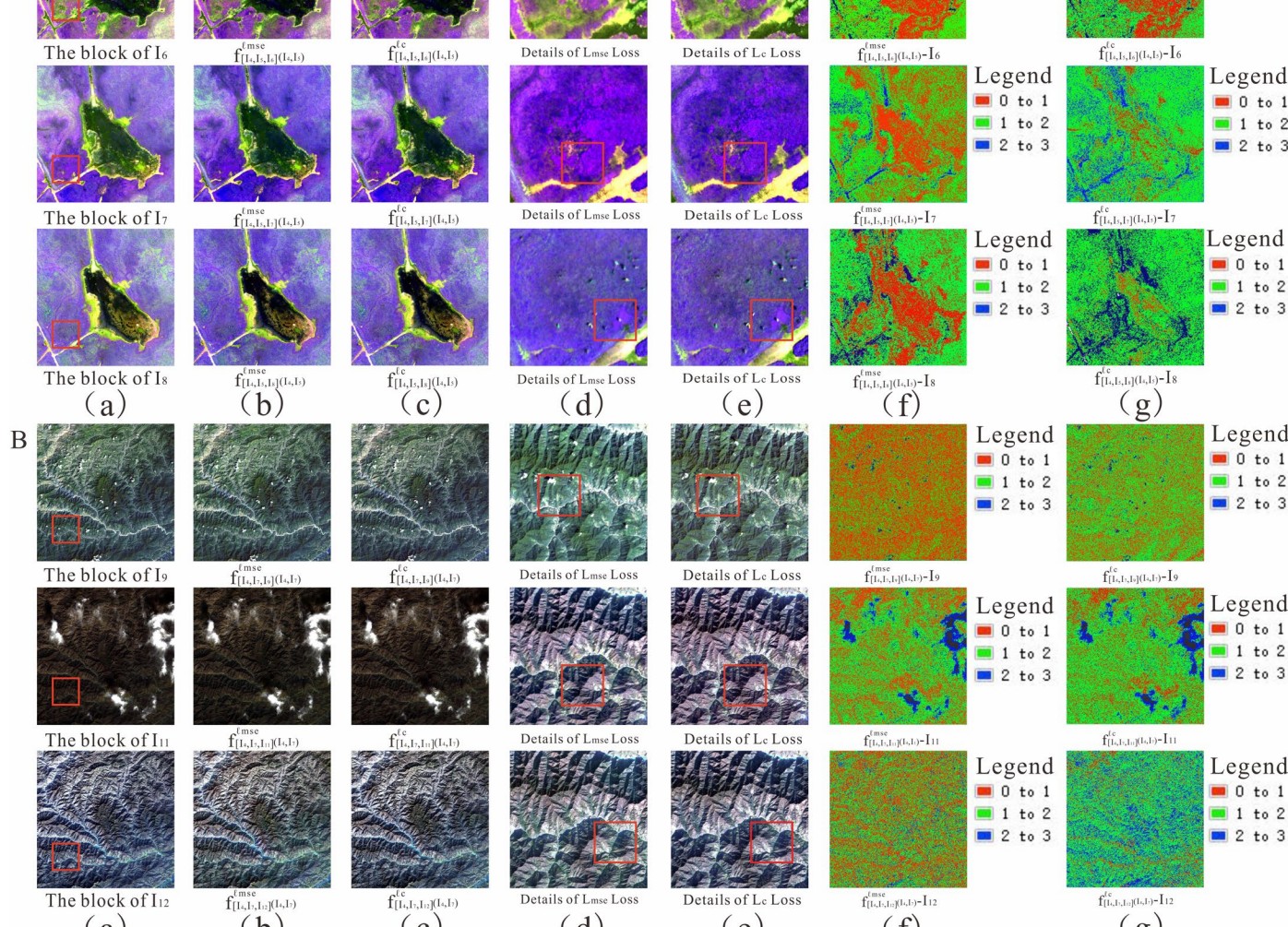

**Figure 7.** Visual effect, detailed information, and pixel error between block interpolated result and reference block using (**A**) UAV and (**B**) Landsat-8 datasets with (**b**, **d**, **f**) $\ell_{mse}$ loss and (**c**, **e**, **g**) $\ell_c$ loss.

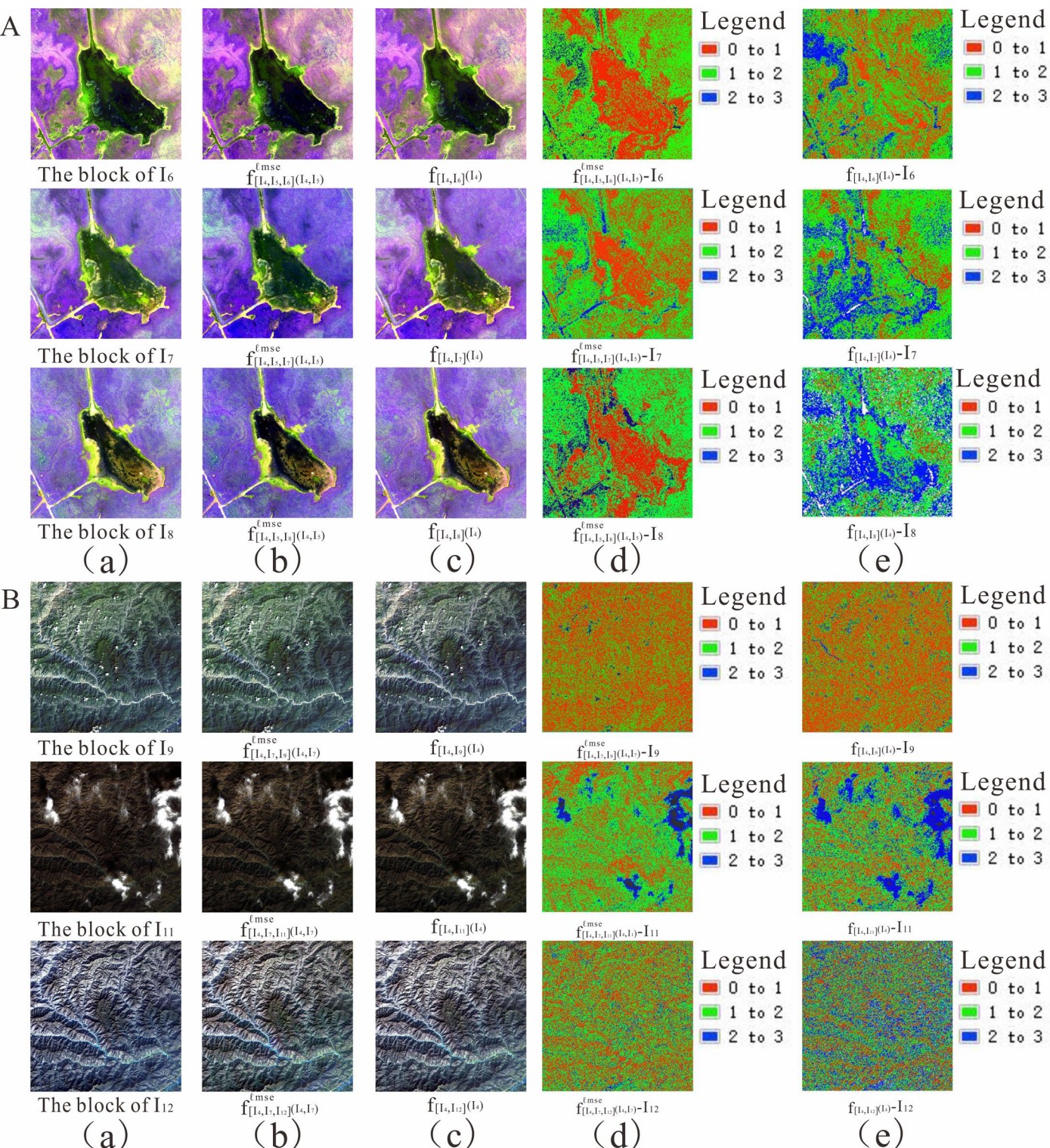

**Figure 8.** Visual effect and pixel error between block interpolated result and reference block using (**A**) UAV and (**B**) Landsat-8 datasets with (**b**, **d**) our proposed method and (**c**, **e**) the method of Meyer et al.

Table 5 shows the quantitative evaluation indicator between the scene interpolated result and reference scene image in the second set of the experiment. The table shows that the entropy value produced by using $\ell_c$ loss was higher than that using $\ell_{mse}$ loss, and the RMSE value produced by using $\ell_{mse}$ loss was lower than that using $\ell_c$ loss. Figures 9 and 10 show the visual effect and pixel error between the scene interpolated result and reference scene image using different loss functions and illustrates that the scene interpolated result using $\ell_{mse}$ loss and $\ell_c$ loss was close to the reference scene image. Figures 11–13 show

the spectral curves between the scene interpolated result and reference scene image using different loss functions at different coordinates (vegetation, pond, ditches, and lake) from June to August 2019 and illustrates that using $\ell_{mse}$ loss and $\ell_c$ loss could maintain better spectral features between the scene interpolated result and reference scene image.

**Table 5.** Quantitative evaluation between scene interpolated result and reference scene image.

| Interpolated Results (Scene) | Reference Images | Entropy | RMSE (Pixel) |
|---|---|---|---|
| $f^{\ell_{mse}}_{[I_4,I_5,I_6]}(I'_4,I'_5)$ | $I_6$ | 3.650 | 1.124 |
| $f^{\ell_c}_{[I_4,I_5,I_6]}(I'_4,I'_5)$ | | 3.656 | 1.163 |
| $f^{\ell_{mse}}_{[I_4,I_5,I_7]}(I'_4,I'_5)$ | $I_7$ | 3.346 | 1.017 |
| $f^{\ell_c}_{[I_4,I_5,I_7]}(I'_4,I'_5)$ | | 3.346 | 1.381 |
| $f^{\ell_{mse}}_{[I_4,I_5,I_8]}(I'_4,I'_5)$ | $I_8$ | 3.494 | 1.210 |
| $f^{\ell_c}_{[I_4,I_5,I_8]}(I'_4,I'_5)$ | | 3.506 | 1.550 |

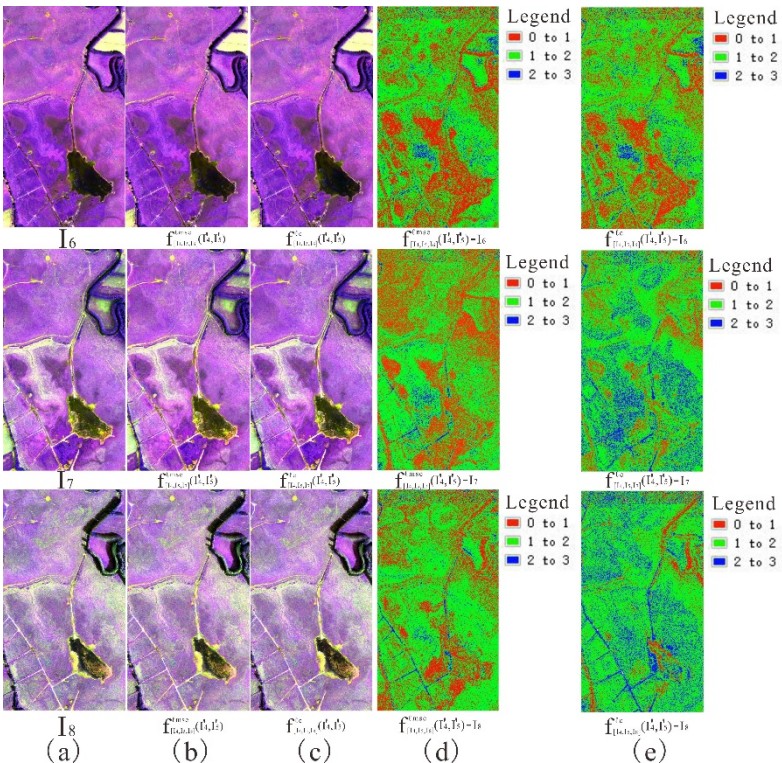

**Figure 9.** Visual effect and pixel error between scene interpolated result and reference scene image using (**a**) Initial image (**b**,**d**) $\ell_{mse}$ loss and (**c**,**e**) $\ell_c$ loss (1, 2, and 3 band composite).

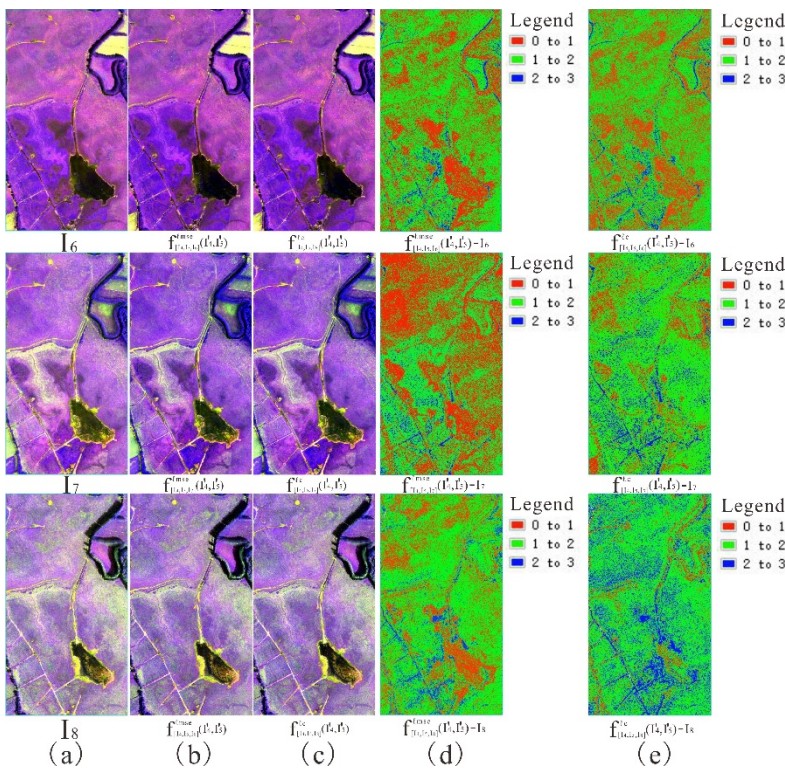

**Figure 10.** Visual effect and pixel error between scene interpolated result and reference scene image using (**a**) Initial image (**b**,**d**) $\ell_{mse}$ loss and (**c**,**e**) $\ell_c$ loss (1, 2, and 4 band composite).

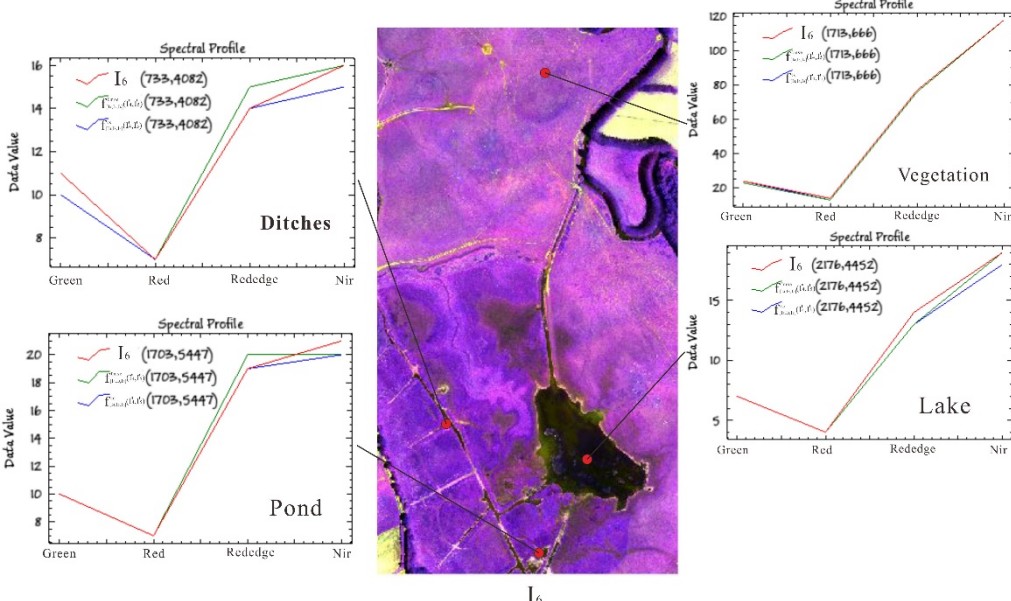

**Figure 11.** Spectral curves between scene interpolated result and reference scene image using different loss functions at different coordinates in June 2019.

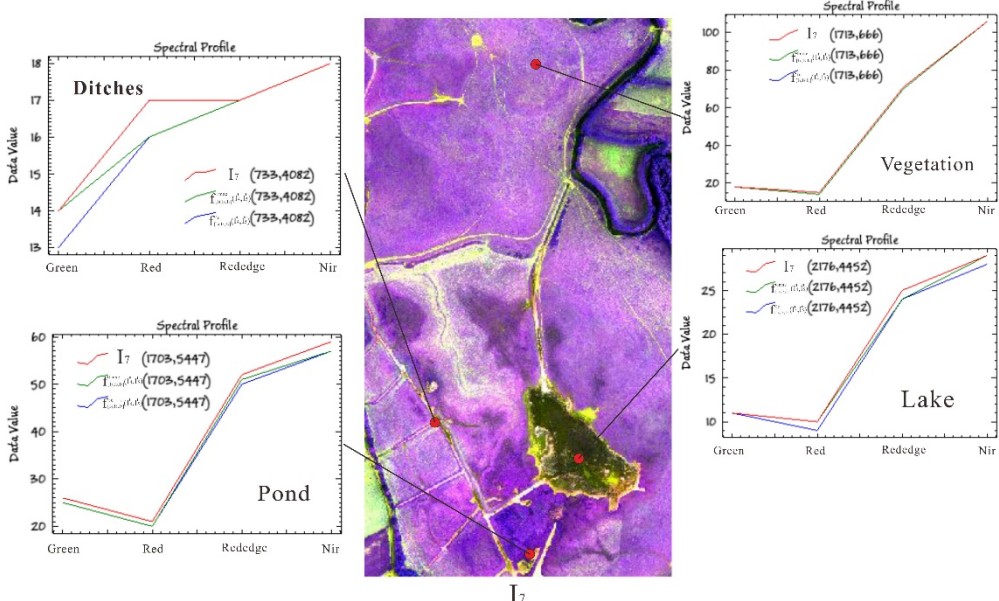

**Figure 12.** Spectral curves between scene interpolated result and reference scene image using different loss functions at different coordinates in July 2019.

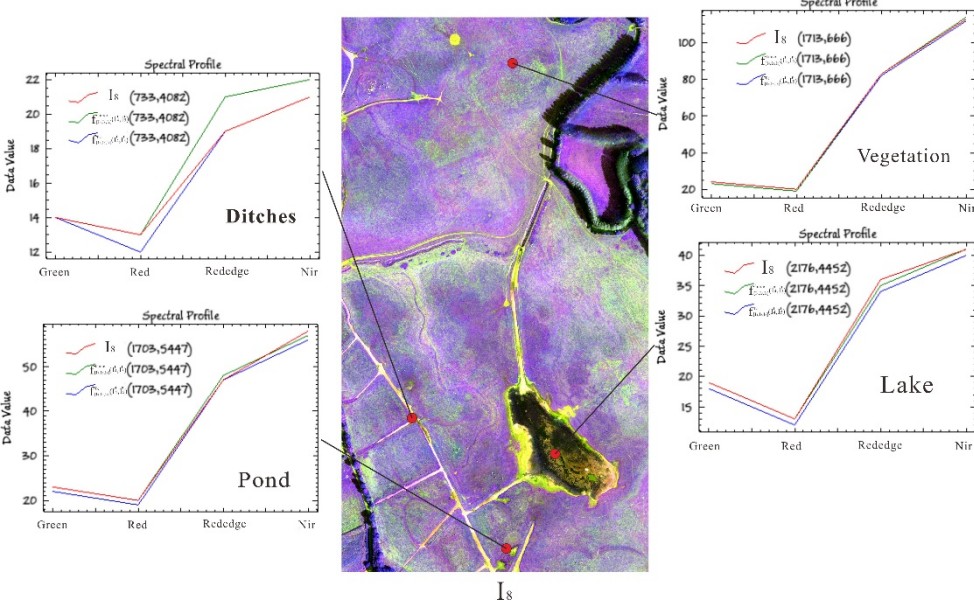

**Figure 13.** Spectral curves between scene interpolated result and reference scene image using different loss functions at different coordinates in August 2019.

In the third set of the experiment, $\ell_c$ loss was used to generate a mapping model. There were no reference images in this experiment. Figure 14 shows the interpolated results of UAV images using $\ell_c$ loss from 2017 to 2019 and the visual effects of three-level interpolation according to the interpolation strategy in Table 2. It appears that as the level of interpolation increased, the spectral features of the interpolated result became worse. This may be caused by the propagation of pixel error as the level of interpolation increased.

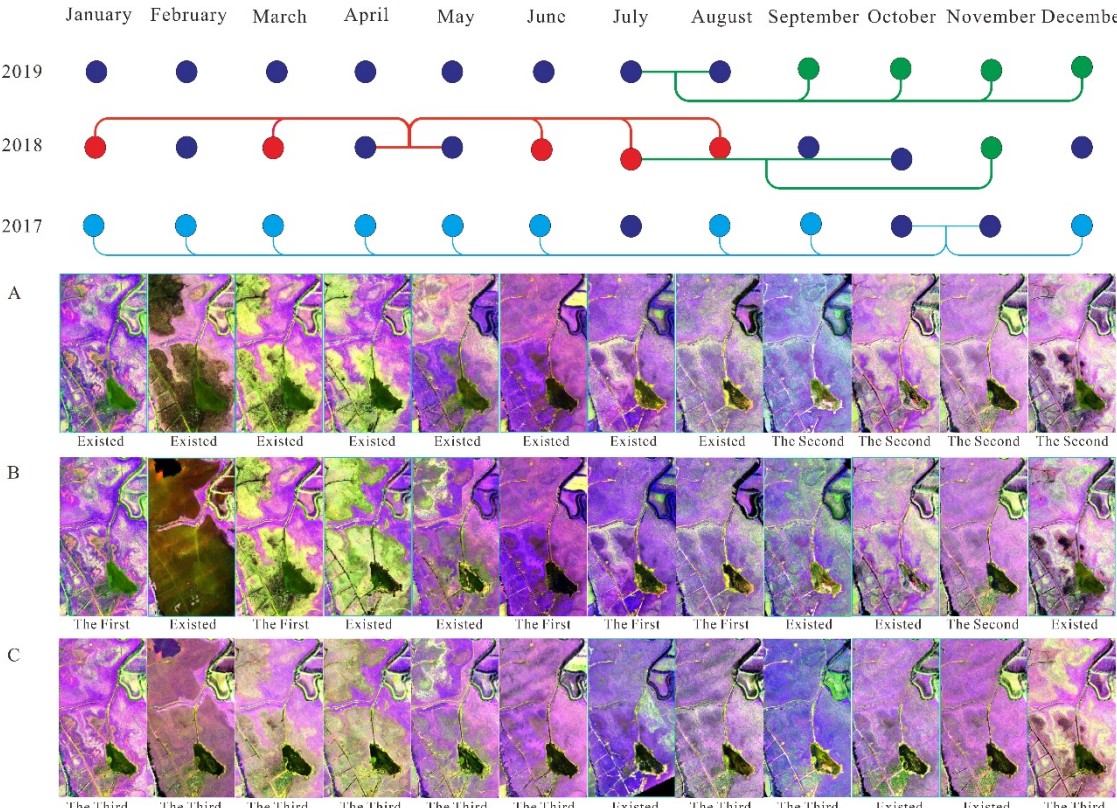

**Figure 14.** Interpolated result of UAV images using $\ell_c$ loss from 2017 to 2019 according to interpolation strategy in Table 2: existing images and interpolated results in (**A**) 2019 sequence, (**B**) 2018 sequence, and (**C**) 2017 sequence.

## 5. Discussion

### 5.1. Stacked Convolution Layers

We observed that the number of stacked convolution layers had an impact on the interpolated result, and conducted a visual comparison between the block interpolated result and reference block using different stacked convolution layers. We selected 1 × 1, 2 × 2, and 3 × 3 stacked convolution layers to train the proposed separable convolution network. Table 6 shows the quantitative evaluation indicator and Figure 15 the visual effect and pixel error between the block interpolated results and reference block using different stacked convolution layers.

**Table 6.** Quantitative evaluation between block interpolated result and reference block using different stacked convolution layers.

| Stacked Numbers | Interpolated Results (Block) | Reference Blocks | RMSE (Pixel) |
|:---:|:---:|:---:|:---:|
| 1 × 1 | $f^{\ell_c}_{[I_4,I_5,I_6]}(I_4,I_5)$ | $I_6$ | 1.375 |
| | $f^{\ell_c}_{[I_4,I_5,I_7]}(I_4,I_5)$ | $I_7$ | 1.556 |
| | $f^{\ell_c}_{[I_4,I_5,I_8]}(I_4,I_5)$ | $I_8$ | 1.666 |
| 2 × 2 | $f^{\ell_c}_{[I_4,I_5,I_6]}(I_4,I_5)$ | $I_6$ | 1.258 |
| | $f^{\ell_c}_{[I_4,I_5,I_7]}(I_4,I_5)$ | $I_7$ | 1.431 |
| | $f^{\ell_c}_{[I_4,I_5,I_8]}(I_4,I_5)$ | $I_8$ | 1.465 |
| 3 × 3 | $f^{\ell_c}_{[I_4,I_5,I_6]}(I_4,I_5)$ | $I_6$ | 1.077 |
| | $f^{\ell_c}_{[I_4,I_5,I_7]}(I_4,I_5)$ | $I_7$ | 1.294 |
| | $f^{\ell_c}_{[I_4,I_5,I_8]}(I_4,I_5)$ | $I_8$ | 1.429 |

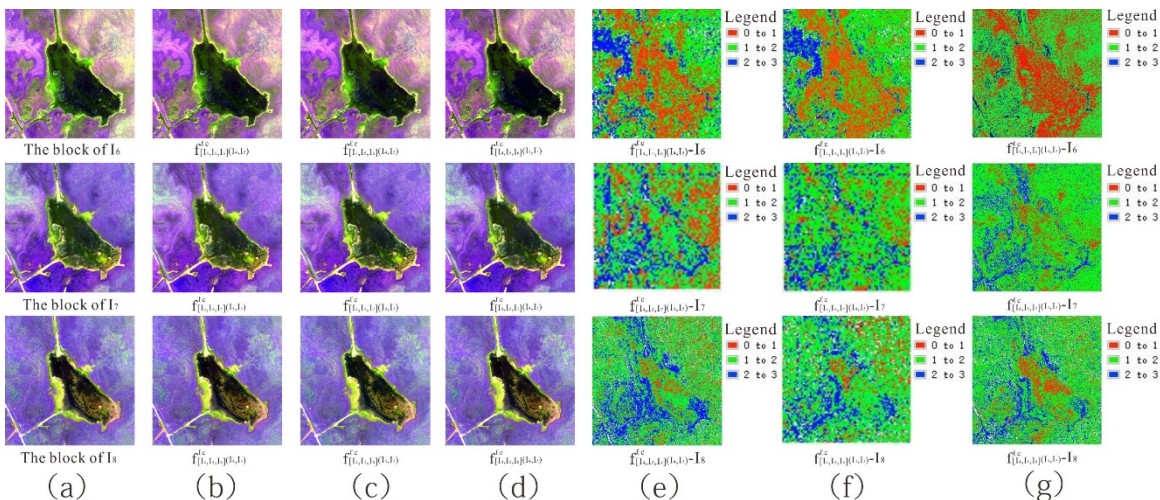

**Figure 15.** (**a**) Initial image (**b**–**d**) Visual effect and (**e**–**g**) pixel error between block interpolated results and reference block using stacks of 1 × 1, 2 × 2, and 3 × 3 convolution layers.

### 5.2. Pooling Type

We observed that the pooling type of the network model had an impact on the interpolated result, and conducted a visual comparison between the block interpolated result and reference block using different pooling types. We selected average pooling and maximum pooling to train the proposed separable convolution network. Table 7 shows the quantitative evaluation indicator and Figure 16 visual effect and pixel error between the block interpolated result and reference block using different pooling types.

**Table 7.** Quantitative evaluation between block interpolated result and reference block using different pooling types.

| Pooling Type | Interpolated Results (Block) | Reference Blocks | RMSE (Pixel) |
|---|---|---|---|
| Average pooling | $f^{\ell c}_{[I_4,I_5,I_6]}{}^{(I_4,I_5)}$ | $I_6$ | 1.326 |
| | $f^{\ell c}_{[I_4,I_5,I_7]}{}^{(I_4,I_5)}$ | $I_7$ | 1.492 |
| | $f^{\ell c}_{[I_4,I_5,I_8]}{}^{(I_4,I_5)}$ | $I_8$ | 1.700 |
| Maximum pooling | $f^{\ell c}_{[I_4,I_5,I_6]}{}^{(I_4,I_5)}$ | $I_6$ | 1.077 |
| | $f^{\ell c}_{[I_4,I_5,I_7]}{}^{(I_4,I_5)}$ | $I_7$ | 1.294 |
| | $f^{\ell c}_{[I_4,I_5,I_8]}{}^{(I_4,I_5)}$ | $I_8$ | 1.429 |

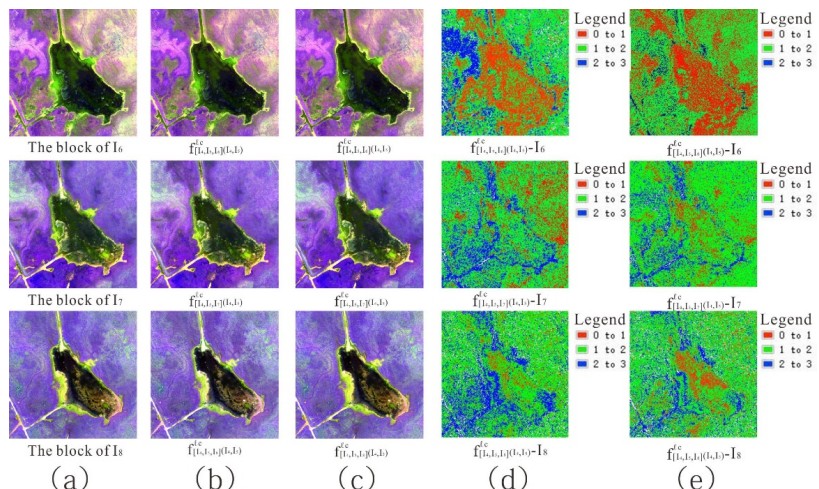

**Figure 16.** Visual effect and pixel error between block interpolated result and reference block using (**a**) Initial image (**b**,**d**) average pooling and (**c**,**e**) maximum pooling.

### 5.3. Temporal Gap between Testing Blocks and Model Requirements

We observed that for a given model, the testing block needed to meet certain requirements. What would happen if a temporal gap existed between testing blocks and those requirements? We conducted a visual comparison between the block interpolated result and reference block when a temporal gap existed between testing blocks and model requirements. We selected several testing blocks (April, May 2018; April, October 2018; and April, December 2018) to test the learned mapping model ($f^{\ell c}_{[I_4,I_5,I_6]}$, $f^{\ell c}_{[I_4,I_5,I_7]}$ and $f^{\ell c}_{[I_4,I_5,I_8]}$); the model required that testing blocks should be acquired in April and May. Table 8 shows the quantitative evaluation indicator and Figure 17 the visual effect and pixel error between the block interpolated result and the reference block using different testing block pairs.

**Table 8.** Quantitative evaluation between block interpolated result and reference block using different testing block pairs.

| Testing Image Date | Interpolated Results (Block) | Reference Blocks | RMSE (Pixel) |
|---|---|---|---|
| April, May 2018 | $f^{\ell c}_{[I_4,I_5,I_6]}(I_4,I_5)$ | | 1.341 |
| April, October 2018 | $f^{\ell c}_{[I_4,I_5,I_6]}(I_4,I_{10})$ | $I_6$ | 3.912 |
| April, December 2018 | $f^{\ell c}_{[I_4,I_5,I_6]}(I_4,I_{12})$ | | 3.989 |
| April, May 2018 | $f^{\ell c}_{[I_4,I_5,I_7]}(I_4,I_5)$ | | 1.498 |
| April, October 2018 | $f^{\ell c}_{[I_4,I_5,I_7]}(I_4,I_{10})$ | $I_7$ | 5.096 |
| April, December 2018 | $f^{\ell c}_{[I_4,I_5,I_7]}(I_4,I_{12})$ | | 5.271 |
| April, May 2018 | $f^{\ell c}_{[I_4,I_5,I_8]}(I_4,I_5)$ | | 1.653 |
| April, October 2018 | $f^{\ell c}_{[I_4,I_5,I_8]}(I_4,I_{10})$ | $I_8$ | 5.313 |
| April, December 2018 | $f^{\ell c}_{[I_4,I_5,I_8]}(I_4,I_{12})$ | | 5.568 |

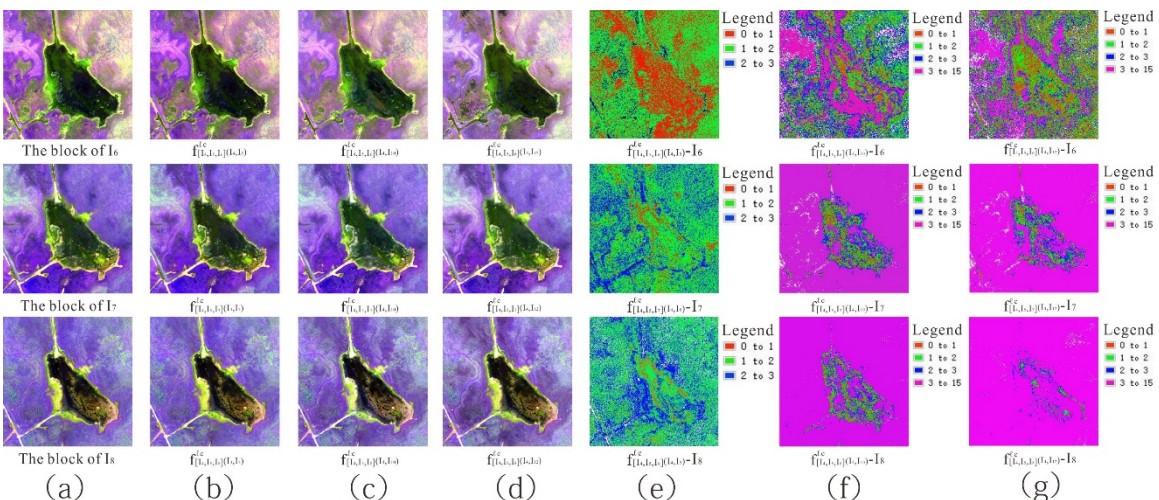

**Figure 17.** (**a**) Initial image (**b**–**d**) Visual effect and (**e**–**g**) pixel error between block interpolated result and reference block using testing block pairs April–May, April–October, and April–December 2018.

### 5.4. Separable Convolution Kernel Size

We observed that the separable convolution kernel size of the network model had an impact on the interpolated result, and conducted a visual comparison between the block interpolated result and reference block using different separable convolution kernel sizes. We selected separable convolution kernels with sizes of 11, 13, and 15 to train the proposed separable convolution network. Table 9 shows the quantitative evaluation indicator

and Figure 18 the visual effect and pixel error between the block interpolated result and reference block using different separable convolution kernel sizes.

**Table 9.** Quantitative evaluation between block interpolated result and reference block using different separable convolution kernel sizes.

| Kernel Size | Interpolated Results (Block) | Reference Blocks | RMSE (Pixel) |
|---|---|---|---|
| 11 | $f^{\ell c}_{[I_4,I_5,I_6]}(I_4,I_5)$ | $I_6$ | 1.077 |
|  | $f^{\ell c}_{[I_4,I_5,I_7]}(I_4,I_5)$ | $I_7$ | 1.294 |
|  | $f^{\ell c}_{[I_4,I_5,I_8]}(I_4,I_5)$ | $I_8$ | 1.429 |
| 13 | $f^{\ell c}_{[I_4,I_5,I_6]}(I_4,I_5)$ | $I_6$ | 1.178 |
|  | $f^{\ell c}_{[I_4,I_5,I_7]}(I_4,I_5)$ | $I_7$ | 1.446 |
|  | $f^{\ell c}_{[I_4,I_5,I_8]}(I_4,I_5)$ | $I_8$ | 1.765 |
| 15 | $f^{\ell c}_{[I_4,I_5,I_6]}(I_4,I_5)$ | $I_6$ | 1.217 |
|  | $f^{\ell c}_{[I_4,I_5,I_7]}(I_4,I_5)$ | $I_7$ | 1.656 |
|  | $f^{\ell c}_{[I_4,I_5,I_8]}(I_4,I_5)$ | $I_8$ | 1.868 |

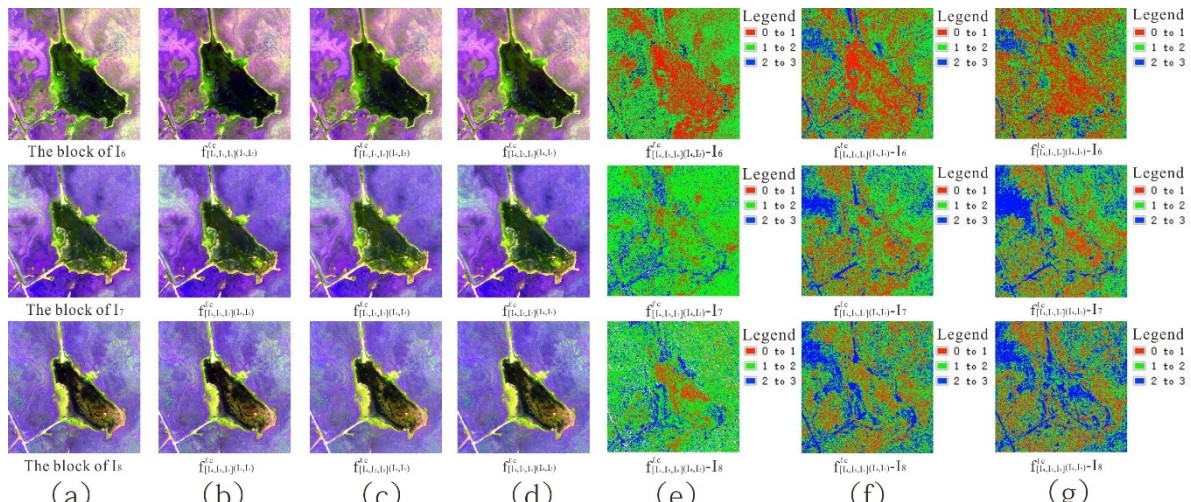

**Figure 18.** (**a**) Initial image (**b**–**d**) Visual effect and (**e**–**g**) pixel error between block interpolated result and reference block using separable convolution kernel sizes 11, 13, and 15.

## 6. Conclusions

The paper presents a remote-sensing sequence image interpolation approach that can transform spectral mapping estimation and pixel synthesis into an easier process of using a separable convolution network to estimate spatially adaptive separable 1D convolution kernels. The conclusions of this paper can be summarized as follows:

(1) The proposed separable convolution network model provides a new method of interpolating remote-sensing images, especially for high-spatial-resolution images. The model can better capture and simulate complex and diverse nonlinear spectral transformation between different temporal images, and get better-interpolated images based on the model.

(2) Using $\ell_c$ loss can produce clearer images in the separable convolutional network compared to $\ell_{mse}$ loss. Using 3 × 3 convolutional layers with ReLu, max pooling, and separable convolution kernel of size 11 led to better-interpolated results in the separable con-

volitional network. Experiments showed that the proposed separable convolution network could be used to get interpolated images to fill in missing areas of sequence images, and produce full remote-sensing sequence images.

The limitation of this method is that the proposed separable convolutional network in this paper can only perform single-scene interpolation, and the quality of the interpolated result depends heavily on the reference image. In future work, the proposed method will be improved from single-scene to multi-scene interpolation using time as a variable to reduce the dependency of interpolation results on the reference image.

**Author Contributions:** Conceptualization, P.T.; Funding acquisition, Z.Z.; Methodology, X.J.; Supervision, P.T.; Writing—original draft, X.J.; Writing—review and editing, T.H., T.C., and E.G.A.-V. All authors have read and agreed to the published version of the manuscript.

**Funding:** This work was supported by the Strategic Priority Research Program of the Chinese Academy of Sciences (grant no. XDA19080301) and the National Natural Science Foundation of China (grant no. 41701399).

**Data Availability Statement:** Data sharing is not applicable to this article.

**Conflicts of Interest:** The authors declare no conflicts of interest.

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
