# Peer review of "Sequence Image Interpolation via Separable Convolution Network"

_remotesensing, doi:10.3390/rs13020296_

Round 1
Reviewer 1 Report
- Authors presented a deep learning method to learn the complex mapping of interpolating the intermediate image from the front and back images and then performs interpolation based on this mapping. The obtained numerical results of simulations verify the advantages of proposed network model: well focusing performance and smaller amount of computation compared with the conventional methods.
- Authors make extensive revision of world bibliography concerning different methods of processing the remote sensing time series data, convolutional networks, pattern recognition an machine learning.
- The architecture of theoretical model after making corrections is properly described in Section 3 and depicted in Figure 3. Also all presented in this paper mathematical formulas and used symbols are now correctly edited and defined.
- The obtained experimental results (root mean square error and entropy function) are properly shown in Tables 3-8 and depicted in Figures 4-17 with the correct comments.
- Sections 4 and 5 have been significantly revised and expanded. In additional Tables 6-8 and Figures 15-17 shown the results of quantitative evaluation and visual effect between the block interpolated result and reference block using different stacked convolution layers, different pooling type of network model and different testing block pairs.
- In this revision version the additional comments added to Figures 4-8 and 14-17 help us to better interpret the obtained experimental results.
- The performed calculations confirmed the effectiveness of proposed model for the tested data sets.
Author Response
Thank you for your high evaluation of my submitted article!
Reviewer 2 Report
The edits applied by the authors are not sufficient in my opinion. There is still a big list of grammatical and scientific corrections. The literature review is not acceptable to me. The method doesn't seem acceptable in my opinion.
Some of my comments:
Line 14: Earth. Make E Capital.
Line 17: “the intermediate image” -> “an intermediate image”
Line 20: “ the input sequence image”->” input sequence images” Where does ‘the’ refer to?
Line 21: “ performed”-> “, which were performed…, show that”
Line 42: Remote sensing->”Remote-sensing”
Line 45: “it is difficult to consider the interpolation of 2D images in the way of 1D data interpolation.” ->” it is difficult to interpolate 2D images in the same way that an interpolation is applied on 1D data.”
Line 46: “The simplest is”Simplest what? There should be a word after simplest like: The simplest interpolation that could be applied on 2D …?
Line 46-47 needs a rewrite. It doesn’t give any understandable information.
Line 51: “What the difference is that video” Please rewrite.
Line 53: “In the following paper,” which paper?
Line 53: “frame and scene” Why you have used these two terms interchangeably since they are not the same.
Line 54: “Nikolas’s method [6] employs a deep full”->It has more than one author. Also the sentence is not acceptable.
Line 60-63 must be rewritten.
Line 64: Literature review is not sufficient.
Line 68: How this is a contribution?
Line 71: Very weak English sentence. The contribution is also very vague to me.
Line 79: Very peculiar use of “And”
Line 88: “to their known data selection” What does this mean?
Line 87-92: Punctuation needs fixes almost anywhere.
Line 93:” In the case of the former methods,” Very strange opening. It is also repeated in line 101.
Author Response
Dear Reviewer,
Thanks a lot for that you reviewed our manuscript and made comments to it. We appreciate you very much for your giving such detailed annotations on our manuscript and so many valuable revision suggestions to us. I have conducted English Editing on MDPI and have carefully revised my manuscript according to your suggestions. All the changes have marked in red color. We hope our revision can be accepted.

Reviewer 3 Report
This paper proposes a deep learning-based frame prediction scheme to estimate remote sensing satellite images. To estimate missing information, the author used a kernel-based frame interpolation scheme. The paper is well-organized. However, there are a few issues to discuss.
1. Reconstructing time-series data using separable convolution is the well-known existing approach. What is a major novelty in this paper? The use of the normalization layer is a general technique used in many papers. It seems that the overall system of the proposed network is similar to the method proposed by [26]. It seems that the performance improvement compared to [26] comes from the use of more information than [26]. This cannot be a novelty of the proposed method.
2. Do you have a special reason to constrain the size of the separable convolution up to 11? It seems that a large kernel shows better performance in the general cases.
3. In terms of visual quality, perceptual loss produces better output than MSE loss. However, this also generates incorrect structures and fine textures. Are there any side effects of using perceptual loss?
Author Response

(The authors gave the same response as above.)

Round 2
Reviewer 3 Report
The paper is greatly improved, and the authors clearly respond most of concerns. I would like to recommend this paper to the publication of this journal.
Author Response
Dear reviewer,
Thank you for your high evaluation of my submitted article!
This manuscript is a resubmission of an earlier submission. The following is a list of the peer review reports and author responses from that submission.
Round 1
Reviewer 1 Report
- In this paper the Authors presented a deep learning method to learn the complex mapping of interpolating the intermediate image from the front and back images and then performs interpolation based on this mapping. The obtained numerical results of simulations verify the advantages of proposed network model: well focusing performance and smaller amount of computation compared with the conventional methods.
- In Introduction (29-74) and Related studies (76-116) the Authors make revision of world bibliography concerning different methods of processing the remote sensing time series data, convolutional networks, pattern recognition an machine learning.
- In my opinion, considering methods and algorithms useful in processing and recognition different kind of images, the following articles concerning the similar problem of images recognition based on the measured signal parameters are also supposed to be listed in References:
- Matuszewski, J.; Pietrow, D. Recognition of electromagnetic sources with the use of deep neural networks. XII Conference on Reconnaissance and Electronic Warfare Systems, Ołtarzew, Poland, 19-21.11.2018, SPIE 11055, 110550D. 2019, DOI: 10.1117/12.2524536.
- Chen, Hongyi; Zhang, Fan; Tang, Bo; et al., Slim and Efficient Neural Network Design for Recourse-Constrained SAR Target Recognition, REMOTE SENSING, Vol. 10, Issue: 10, Article Number: 1618, Published: OCT 2018.
- Xu, Wang; Chen, Renwen; Huang, Bin; et al., Single Image Super-Resolution Based on Global Dense Feature Fusion Convolutional Network, SENSORS, Vol.: 19, Issue: 2, Article Number: 316, Published: JAN 2 2019.
- The architecture of theoretical model is correctly described in Section 3 and depicted in Figure 1. All presented in this paper mathematical formulas and used symbols are correctly
- The successive steps of training, testing and validation for proposed method with appropriate mathematical formula and computing examples are well illustrated in the appropriate Figures (2-6).
- The experimental results (mean square error and entropy function) are shown in the Tables 2-3 and depicted in Figures 4-13 with appropriate comments.
- Remarks:
- The abbreviations HANTS (line 91) and VGG (line 256) are deciphered.
- Description of all mathematical symbols used in the text should be written with the same style (italic font) just like in the formulas, for example: b1, K1, K2, D(I), i,
- It is very difficult to interpret the results of the calculations shown in Figures 7-13. Could the authors add 1-2 sentences to each of these figures, which the image most closely reflects the features of the real image?
- The simulation tests carried out confirmed the usefulness of proposed by Authors method for set of data taken to calculations. The proposed by the Authors model of separable convolutional network improve the operation speed and effectiveness of images interpolation.
Reviewer 2 Report
This paper was about temporal image interpolation by using convolutional neural networks. The topic seems interesting specially when datasets lack some frames from specific seasonal spots. The overall scientific tone of this work was acceptable to me. The methodology was partly described, and results were to some extend demonstrated; however, I have several major concerns:
- suitability of the method, since the author did not quantitatively compare his proposed method with other methods. In one part of text it is stated that linear interpolation is unable to do the job, but no real demonstration of a linear or a polynomial interpolation is stated. Different lost functions are compared in figure 7 e.g., however, the same figure could be compiled to show the superiority of this method in comparison to other interpolation methods such as bilinear or bicubic. I didn’t find anywhere in the text why a polynomial interpolation e.g. is unable to perform the interpolation task.
- The way that the author considers training and test datasets seems concerning to me, since the training and test datasets are from same images. It was a better strategy if they used some images for training and then applied the trained network on completely different images. It is also not clear how the time intervals are taken into account in different classifier, meaning that a network that is trained for June-April could not possibly be employed for other time intervals.
- The structure of the network is not well described. Figure 1 needs to be improved to exactly show the network’s structure.
- Novelty of the approach is not stated enough and a comparison to other works is a must.
- The article needs intensive English editing in my opinion. I strongly recommend the authors to improve this aspect of their work, since the current status contains many ambiguous texts.
By reading the text thoroughly I have also found the following problems that needs to be addressed:
- Line 18: “Front and back” images are usually referred to multi camera settings. It is better to change it with a more suitable term.
- Line 19: “the separable”->” a separable”
- Line 20: “the 2D kernels”->”2D kernels”
- Line 34: Indices of references should start at 1
- Line 36: add a suitable reference.
- Structure of introduction is weak. It should starts with general definition of the proble, followed by a literature review, and then a short presentation of achieving points of the article. Please update.
- Line 38 and 48: “This paper…” This should be moved to the last part of introduction.
- Line 50-52: Please quantitatively compare your model with simple linear method to show that a linear blending is unable to address this problem well.
- Line 51: Which surface features?
- Line 68: Please state how much memory is exactly required to run with 2D kernels, 10X improvement is very vague.
- Line 71:74 very complex, please reorganize
- Line 76: “Data reconstruction” or “data interpolation”?
- Line 77: “Spatial neighborhood data” please clarify what you mean?
- Line 81: All abbreviations should be corrected such as expansion comes outside parenthesis and then abbreviation is stated inside a parenthesis.
- Line 82: Abbreviation problem.
- Line 90: “[35] use”-> “[35] used”
- Line 94: Should be moved to the discussion.
- Line 97 to discussion
- Line 108-116 to discussion
- Figure 1 needs major improvement
- Line 143: Describe better
- Line 148-152 -> discussion
- Lines 163-164 please rewrite.
- Figure 2: Please follow standards of the journal for this figure.
- Line 176-186: How many networks have you trained?
- Line 189: Please rewrite.
- Line 200-202: rewrite
- Line 208: something is missing: “is mainly to”-> probably “is mainly designed to”
- Line 209: “Illustrate” is inappropriate for “table” English edit.
- Line 252: what do you mean by feature extraction? Equation 3 is vague to me .
- What is the benefit of entropy over MSE please state. MSE units are missing in the tables.
- What you didn’t use root mean square error (RMSE)?
- How different loss functions were considered into your implantation?
- Line 365: Please remove questions and follow a standard discussion format.
In overall I recommend the journal to accept this article after a major revision.
Reviewer 3 Report
This paper proposes the deep learning based method to reconstruct remote sensing satellite images. The proposed method used kernel-based frame interpolation scheme for the reconstruction of images. However, there are several issues to be future considered.
- Most of all, frame prediction based on past frames using separable convolution is not a new idea. This kind of application is already proposed. What is main contribution of this paper? It seems hard to find any novelty from the paper.
- There are similar work to interpolate the intermediate frame using reference image [1]. Please include the following reference paper, and compare the proposed method with it.
[1] Meyer, Simone, et al. "Deep video color propagation." arXiv preprint arXiv:1808.03232 (2018).
- The estimation of unknown frame in the proposed method seems more similar to the extrapolation, rather than interpolation. Is it okay to say interpolation in the paper?
- Using the separable convolution, it can only predict a frame at a specific time, which is one of the weaknesses for the separable convolution. In testing strategy section, however, many frames which have different time are predicted. How is it possible to predict by one network? Is it predicted by many different networks? Please describe your testing process more clearly.
- In the paper, what is the meaning of "reference image" exactly? Please define it more carefully and clearly.
- To verify superiority of the proposed approach, comparison with other methods should be added.
- Also, it is highly recommended to verify superiority of the proposed method in other datasets.
Reviewer 4 Report
The paper was well written.
I would recommend adding more detail on the limitations of using your method with respect to the sensitivity of time between images actually collected. In other words, the greater the time distance between the two images used for interpolation, the greater the chance for error in the interpolated image. So, discuss what your recommendation for "maximum time between images" would be in using your method. (i.e. discuss the limitations of your method)
Second, I would recommend including a "Future Works" paragraph at the end of the paper.
Round 2
Reviewer 2 Report
My comments are in the attached file. More improvements are required.

Reviewer 3 Report
The revised paper has addressed most of my concerns. However, there are still some issues to discuss.
- The novelty of this method is still not satisfactory. The basic concept of the proposed method seems to be similar to the method in Meyer et al.
- Evaluation datasets are not enough to prove the superiority of the proposed method. In particular in Table 4, comparisons between the proposed and other methods are just performed for one image.
- Tables for quantitative evaluation are not properly organized.